# Multicentre survey of retinopathy of prematurity in Indonesia

J Edy Siswanto ![ORCID],[1,2] Arend F Bos,[3] Peter H Dijk,[1] Rinawati Rohsiswatmo,[4] Gatot Irawan,[5] Eko Sulistijono,[6] Pertin Sianturi,[7] Dewi A Wisnumurti,[8] Rocky Wilar,[9] Pieter J J Sauer,[1] The IMSROP Study Group

► Additional material is published online only. To view, please visit the journal online (http://dx.doi.org/10.1136/bmjpo-2020-000761).

For numbered affiliations see end of article.

**Correspondence to**
Dr J Edy Siswanto; j.e. siswanto@umcg.nl

## ABSTRACT

**Background** The incidence of retinopathy of prematurity (ROP) is higher in Indonesia than in high-income countries. In order to reduce the incidence of the disease, a protocol on preventing, screening and treating ROP was published in Indonesia in 2010. To assist the practical implementation of the protocol, meetings were held in all Indonesia regions, calling attention to the high incidence of ROP and the methods to reduce it. In addition, national health insurance was introduced in 2014, making ROP screening and treatment accessible to more infants.

**Objective** To evaluate whether the introduction of both the guideline drawing attention to the high incidence of ROP and national health insurance may have influenced the incidence of the disease in Indonesia.

**Setting** Data were collected from 34 hospitals with different levels of care: national referral centres, university-based hospitals, and public and private hospitals.

**Methods** A survey was administered with questions on admission numbers, mortality rates, ROP incidence, and its stages for 2016–2017 in relation to gestational age and birth weight.

**Results** We identified 12 115 eligible infants with a gestational age of less than 34 weeks. Mortality was 24% and any stage ROP 6.7%. The mortality in infants aged less than 28 weeks was 67%, the incidence of all-stage ROP 18% and severe ROP 4%. In the group aged 28–32 weeks, the mortality was 24%, all-stage ROP 7% and severe ROP 4%–5%. Both mortality and the incidence of ROP were highest in university-based hospitals.

**Conclusions** In the 2016–2017 period, the infant mortality rate before 32 weeks of age was higher in Indonesia than in high-income countries, but the incidence of ROP was comparable. This incidence is likely an underestimation due to the high mortality rate. The ROP incidence in 2016–2017 is lower than in surveys conducted before 2015. This decline is likely due to a higher practitioner awareness about ROP and national health insurance implementation in Indonesia.

## What is known about the subject?

► Several studies in the previous decade in Indonesia showed a high incidence of ROP, which is a serious problem, as in other low/middle-income countries.
► The high rate of retinopathy of prematurity (ROP) in Indonesia is likely due to the expansion of neonatal intensive care, whereas there is a lack of awareness of the risks of developing ROP.

## What this study adds?

► The incidence of ROP in Indonesia was lower in 2016–2017 than in surveys conducted before 2015.
► There was a difference in mortality and rate of ROP between the types of hospital. Both mortality and the incidence of ROP were highest in university-based hospitals.
► The lower rate of ROP could be the result of a new national guideline on ROP, an increased awareness following meetings and the introduction of national healthcare.

## INTRODUCTION

Studies on the incidence of retinopathy of prematurity (ROP) during the period 2005–2015 in Indonesia showed a relatively high incidence of ROP.[1 2] ROP was also seen in infants with gestational ages above 32 weeks. In view of these findings, a group of neonatologists and ophthalmologists met in 2009 and 2010 to develop a guideline for the prevention, early detection and management of ROP for neonatologists and ophthalmologists in Indonesia. The national guideline, which was published in 2010, provides advice on how to prevent ROP and how and when to screen newborn infants.[3] It recommends screening all infants born before 34 weeks or with a birth weight of less than 1500 g, as well as infants of higher gestational age and higher birth weight who received oxygen for a prolonged period. Neonatologists and paediatric ophthalmologists realised that the guideline's publication had almost no impact on clinical practice. This was most likely due to a lack of knowledge about ROP among paediatricians and ophthalmologists and due to financial constraints in hospitals.

After 2010, the paediatric and ophthalmologist societies in Indonesia therefore organised several meetings to alert their members to the high incidence of ROP and ways to

prevent the disease and perform screening procedures. In addition, national health insurance was introduced in 2014, thereby increasing opportunities to treat premature infants and to improve ROP screening. Care for preterm infants in Indonesia is provided by two national referral centres, university-based NICUs (neonatal intensive care centres), as well as governmental and private hospitals. There are marked variations among these hospitals, such as patients' socioeconomic and cultural backgrounds and other demographic factors, in terms of their options for caring for sick preterm infants. We do not know whether the differences between hospitals resulted in a different incidence of ROP.

To evaluate whether an awareness of the high incidence of ROP and the introduction of national health insurance may have reduced that incidence, we conducted a survey on the incidence of the disease at Indonesian NICUs and local hospitals in the years 2016–2017. As the incidence may differ between hospitals, we collected data from all levels of hospitals that provide care for sick newborns.

## METHODS
This is a survey: we collected data for the years 2016–2017 in the period from March to November 2019. Paediatricians in 47 hospitals were contacted by email and direct phone calls; 41 were willing to send us the required information. We received responses from 34 hospitals in 17 major provinces of Indonesia—16 teaching hospitals, 2 of which are national referral hospitals for perinatology, and 10 government and 8 private hospitals. The availability of NICU beds varied greatly across between regions because of a lack of trained neonatologists and differences in stakeholder support in the province or district where the paediatricians worked. We approached hospitals offering all levels of neonatal care, located in all the different parts of Indonesia.

The survey asked for the following data: number of inborn preterm infants, number of preterm infants who died in the perinatal period, number of infants screened for ROP, number of infants with ROP and the stage of ROP. For the sake of uniformity and to analyse all the

data, we asked for inborn babies to be further categorised by gestational age and birth weight. For ROP, we used the terms mild ROP (stages 1–2) and severe ROP (stage 3 or higher). We included only inborn infants because important data such as gestational age and complications in pregnancy are often not available for outborn infants. We analysed the university-based hospitals and the district (government)/private hospitals in two separate groups. The data for the two national referral perinatal centres are shown separately. The results are described in frequencies and percentages. No advance statistical test was used for the data analysis.

## PATIENT AND PUBLIC INVOLVEMENT
Neither patients nor the public were involved in the design, execution, reporting or dissemination plans for our research.

## RESULTS
We received data from the two national referral hospitals for perinatology (Harapan Kita Women and Children's Health Centre and RSCM-Ciptomangunkusumo Hospital), 14 university-based hospital NICUs and 18 other hospitals, which included 10 government hospitals and 8 private hospitals (table 1).

In total, we received data on 12 115 infants with a gestational age <34 weeks, 5252 of whom had a birth weight of less than 1500 g. The overall mortality of infants <34 weeks was 24.1%. Almost 37% of surviving infants were screened for ROP; the incidence of all-stage ROP was 6.7%. The highest incidence of both mortality and ROP was found in the university-based hospitals (table 1).

Table 2 shows the data for each hospital or group of hospitals according to gestational age. The overall mortality in the group <28 weeks was 67%, with no differences between hospitals. Sixty-three per cent of surviving infants were screened for ROP. The rate of screening ranged from 93% in one referral hospital to 42% in the 'other hospitals' group. The overall rate of any-stage ROP was 18% and severe ROP 4%. The incidence of both any

**Table 1** Hospital-based ROP data surveillance in Indonesia 2016–2017 (GA ≤34 weeks)

| Type of hospital | Total hospitals | Total infants | Survived | Died n | Died % | Screened n | Screened % | ROP n | ROP % |
|---|---|---|---|---|---|---|---|---|---|
| RSCM/NRH | 1 | 1038 | 890 | 148 | 14.3 | 392 | 44.0 | 7 | 1.8 |
| HKWCH | 1 | 478 | 392 | 86 | 18.0 | 281 | 71.7 | 7 | 2.5 |
| UBH | 14 | 6549 | 4738 | 1811 | 27.7 | 2041 | 43.1 | 197 | 9.7 |
| OH | 18 | 4050 | 3261 | 789 | 19.5 | 711 | 21.8 | 17 | 2.4 |
| Government hospital | 10 | 3020 | 2425 | 595 | 19.7 | 605 | 24.9 | 15 | 2.5 |
| Private hospital | 8 | 1030 | 836 | 194 | 18.8 | 106 | 12.7 | 2 | 1.9 |
| All hospitals | 34 | 12 115 | 9281 | 2834 | 23.4 | 3425 | 36.9 | 228 | 6.7 |

HKWCHC, Harapan Kita Women and Children's Hospital (National Centre for Women and Children's Health); OH, other hospitals; ROP, retinopathy of prematurity; RSCM/NRH, Ciptomangunkusumo Hospital (national referral hospital); UBH, university-based hospital.

**Table 2** (A) ROP incidence in Harapan Kita Women and Children's Hospital, Indonesia based on the gestational age in 2005–2017; (B) ROP incidence in Indonesia based on the gestational age in 2016–2017

| (A) Variable | | Infants GA <28 weeks | | Infants GA 28–32 weeks | | Infants GA >32–34 weeks | |
|---|---|---|---|---|---|---|---|
| | | 2005–2015 | 2016–2017 | 2005–2015 | 2016–2017 | 2005–2015 | 2016–2017 |
| Total (inborn infants) | n | 185 | 54 | 569 | 221 | NA | 203 |
| Died | n | 91 | 27 | 126 | 24 | – | 24 |
| | % | 49 | 50 | 22 | 11 | – | 12 |
| Survived | n | 94 | 27 | 443 | 197 | – | 179 |
| Screened | n | 47 | 25 | 261 | 169 | 156 | 87 |
| Screened/survived | % | 50 | 93 | 59 | 86 | – | 49 |
| No ROP | n | 28 | 23 | 187 | 167 | 126 | 84 |
| ROP 1–2 | n | 9 | 2 | 66 | 1 | 29 | 3 |
| ROP 3–5 | n | 10 | 0 | 8 | 1 | 1 | 0 |
| Prevalence of any ROP | % | 40 | 8 | 28 | 1 | 19 | 3 |
| Prevalence of severe ROP | % | 21 | 0 | 3 | 1 | 1 | 0 |

| (B) Variable | | Infants GA <28 weeks 2016–2017 | | | | | Infants GA 28–32 weeks 2016–2017 | | | | | Infants GA >32–34 weeks 2016–2017 | | | | |
|---|---|---|---|---|---|---|---|---|---|---|---|---|---|---|---|---|
| | | RSCM | HKWCH | UBH | OH | All | RSCM | HKWCH | UBH | OH | All | RSCM | HKWCH | UBH | OH | All |
| Total (inborn infants) | n | 115 | 54 | 903 | 354 | 1426 | 501 | 221 | 2288 | 2044 | 5054 | 422 | 203 | 3358 | 1652 | 5635 |
| Died | n | 87 | 27 | 577 | 259 | 961 | 122 | 24 | 700 | 349 | 1195 | 26 | 24 | 534 | 181 | 765 |
| | % | 76 | 50 | 64 | 73 | 67 | 24 | 11 | 31 | 17 | 24 | 6 | 12 | 16 | 11 | 14 |
| Survived | n | 28 | 27 | 326 | 95 | 465 | 379 | 197 | 1588 | 1695 | 3859 | 396 | 179 | 2824 | 1471 | 4870 |
| Screened | n | 23 | 25 | 204 | 40 | 292 | 187 | 169 | 923 | 315 | 1594 | 182 | 87 | 914 | 356 | 1539 |
| Screened/survived | % | 82 | 93 | 63 | 42 | 63 | 49 | 86 | 58 | 19 | 41 | 46 | 49 | 32 | 24 | 32 |
| No ROP | n | 22 | 23 | 161 | 33 | 239 | 185 | 167 | 751 | 309 | 1478 | 178 | 84 | 441 | 352 | 1480 |
| ROP 1–2 | n | 1 | 2 | 34 | 5 | 42 | 2 | 1 | 94 | 6 | 103 | 2 | 3 | 45 | 2 | 52 |
| ROP 3–5 | n | 0 | 0 | 9 | 2 | 11 | 0 | 1 | 12 | 0 | 13 | 2 | 0 | 3 | 2 | 7 |
| Prevalence of any ROP | % | 4 | 8 | 21 | 18 | 18 | 1 | 1 | 11 | 2 | 7 | 2 | 3 | 5 | 1 | 3.8 |
| Prevalence of severe ROP | % | 0 | 0 | 4 | 5 | 4 | 0 | 1 | 1 | 0 | 1 | 1 | 0 | 0 | 1 | 0.4 |

GA, gestational age; HKWCH, Harapan Kita Women and Children Hospital (National Centre for Women and Children's Health); NA, not available; OH, other hospitals; ROP, retinopathy of prematurity; RSCM/NRH, Ciptomangunkusumo Hospital (national referral hospital); UBH, university-based hospital.

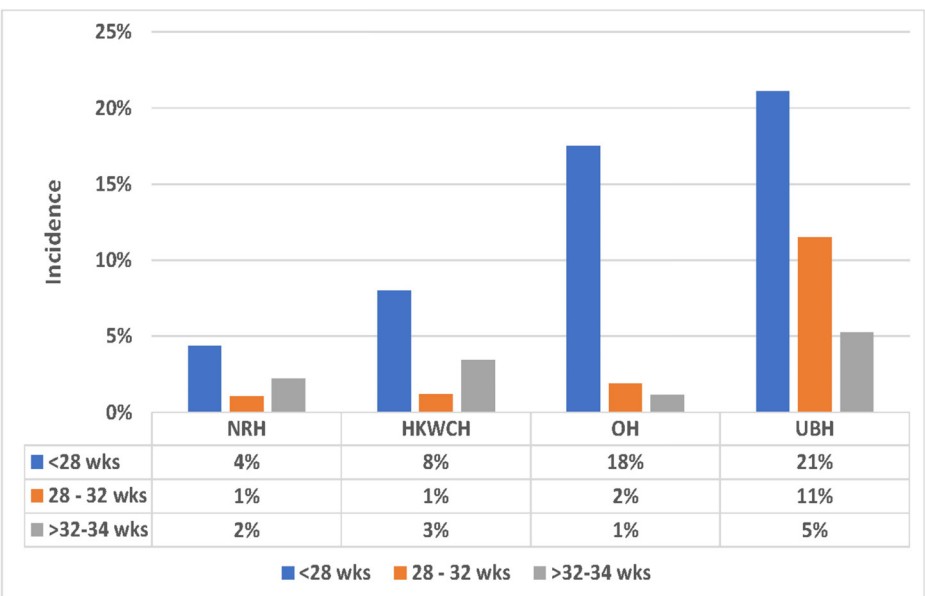

**Figure 1** The incidence of ROP based on gestational age in each group of hospitals in Indonesia 2016–2017. HKWCH, Harapan Kita Women and Children's Hospital (National Centre for Women and Children's Health); NRH, national referral hospital; OH, other hospitals; ROP, retinopathy of prematurity; UBH, university-based hospital; wks, weeks.

stage ROP and severe ROP was higher in the university-based NICUs and other hospitals than in the two national centres (18% and 21% vs 4% and 8%) (figure 1). Severe ROP was not seen in the national centres, but was found in 4% and 5% of infants in the university and other hospitals, respectively. The same trend was found in infants born after 28–32 weeks. Overall mortality in this group was 24%, with the highest mortality in the university hospitals (31%), compared with 11% in one of the national centres. The rate of screening in the group aged 28–32 weeks was lower than in the group aged <28 weeks (41% vs 63%) and ranged from 86% in one national centre to 19% in the 'other hospitals' group. Any stage ROP was found in 7% of surviving infants, with the highest incidence (11%) in university hospitals and 1%–2% in other hospitals. The incidence of severe ROP was 0%–1% in all hospitals. The mortality in the group aged 32–34 weeks was 14% on average, with no important differences between hospitals. The rate of screening in these infants was 32% and the prevalence of any stage ROP was 4%, with the highest incidence in university-based hospitals.

The online supplemental table shows the data for each hospital or group of hospitals according to birth weight. The overall mortality in the <1000 g group was 61%, with no differences between groups of hospitals. Forty-six per cent of surviving infants were screened for ROP; the rate of screening ranged from 88% in one referral hospital to 33% in the 'other hospitals' group. The overall rate of any stage ROP was 18% and severe ROP 3%. The incidence of both any stage ROP and severe ROP was higher in the university-based NICUs and other hospitals than in the two national centres (20% vs 8% and 10%). Severe ROP was not encountered in the national centres but was seen in 3% and 6%, respectively of infants in the university

and other hospitals. The same trend was found in infants born with a birth weight of 1000–1500 g. Overall mortality in this group was 30%, with the highest mortality in the university and other hospitals (30%), compared with 15% in one of the national centres. The rate of screening in the 1000–1500 g group was higher than in the <1000 g group (55% vs 46%), and ranged from 90% in one national centre to 36% in the 'other hospitals' group. Any-stage ROP was found in 8% of surviving infants, with the highest incidence (13%) in the university hospitals and 3% in other hospitals. The incidence of severe ROP was 1% in all hospitals. The mortality in the >1500–2000 g group was 12% on average, with no significant differences between hospitals. The rate of screening in these infants was 33% and the prevalence of any stage ROP was 3%, with the highest incidence in university-based hospitals (5%).

In order to assess whether the screening rate may have affected the incidence of ROP found in our survey, we calculated the incidence of ROP in 13 of the 34 hospitals where at least 80% of infants were screened once or more (table 3). There was no difference in the rate of ROP between these hospitals and the total group. In the <28 weeks group, the ROP rate at all stages was 21% and 5% for severe ROP. In the group aged 28–32 weeks, the incidence of all ROP stages was 6% and severe ROP 1%.

## DISCUSSION

Globally, ROP is one of the hidden causes of morbidity following death-related problems such as respiratory distress (asphyxia), infections and other complications that arise in small, sick preterm infants. Countries with a high rate of premature births will consequently see a higher number of infants affected by ROP. This situation

**Table 3** Incidence of ROP in hospitals with 80% screening coverage in Indonesia, 2016–2017

| Variable | | Birth weight | | | | | Gestational age | | | | |
|---|---|---|---|---|---|---|---|---|---|---|---|
| | | <1000 g | 1000–1500 g | >1500–2000 g | >2000–<2500 g | ≥2500 g | <28 weeks | 28–32 weeks | 32–34 weeks | 34–<37 weeks | ≥37 weeks |
| Total (inborn) | n | 585 | 1541 | 2679 | 4522 | 20572 | 560 | 2144 | 2319 | 3913 | 20963 |
| Died | n | 394 | 449 | 303 | 163 | 266 | 391 | 508 | 244 | 165 | 267 |
| | % | 67 | 29 | 11 | 4 | 1 | 70 | 24 | 11 | 4 | 1 |
| Survived | n | 191 | 1092 | 2376 | 4359 | 20306 | 169 | 1636 | 2075 | 3748 | 20696 |
| Screened | n | 172 | 1040 | 1033 | 1186 | 14 | 168 | 1147 | 926 | 1199 | 5 |
| Screened/survived | % | 90 | 95 | 43 | 27 | 0 | 99 | 70 | 45 | 32 | 0 |
| No ROP | n | 137 | 966 | 978 | 1159 | 14 | 132 | 1074 | 874 | 1169 | 5 |
| ROP 1–2 | n | 28 | 57 | 53 | 27 | 0 | 27 | 62 | 46 | 30 | 0 |
| ROP 3–5 | n | 7 | 17 | 2 | 0 | 0 | 9 | 11 | 6 | 0 | 0 |
| Incidence of any ROP | % | 20 | 7 | 5 | 2 | 0 | 21 | 6 | 6 | 3 | 0 |
| Incidence of severe ROP | % | 4 | 2 | 0 | 0 | 0 | 5 | 1 | 1 | 0 | 0 |

These data come from 13 of 34 hospitals.
ROP, retinopathy of prematurity.

occurs in almost all parts of the world, both in high-income (HIC) and lower/middle-income countries (LMIC).[4]

What has been characterised as the third ROP epidemic has particularly affected countries in Southeast Asia. It is clear that many cases of visual impairment due to ROP are preventable through improved neonatal care, timely retinal examination and appropriate treatment.[5] In this survey, we present data on the incidence and severity of ROP following the implementation of a national guideline for the prevention, screening and treatment of ROP and the introduction of national health insurance in Indonesia, one of the major regions in Southeast Asia.

This survey conducted in 2016–2017 found the incidence of ROP in Indonesia in that period to be much lower than the incidence we found in 2005–2015.[1 2] In that previous period, we found in one NICU a 40% incidence of all-stage ROP in infants below 28 weeks and a 28% incidence in the group aged 28–32 weeks.[1] Other studies from the period 2005–2015 in Indonesia found a ROP incidence of 18%–30% in infants born before 32 weeks and/or with a birth weight of less than 1500 g.[2] Different factors may have contributed to this marked decline in ROP. First, a national guideline for ROP prevention and screening was published in 2010. When we realised that simply publishing this guideline did not change practices in Indonesia, several focus meetings were held across the country to alert practitioners to the high incidence of ROP, the methods to prevent ROP and the importance of screening.[3] In addition, national health insurance was introduced in 2014, thereby improving opportunities for treating and screening preterm infants. We cannot identify which of these factors may have had the greatest impact on the decline in incidence of ROP.

It is difficult to compare the incidence of ROP in Indonesia with the incidence reported in HIC. A recent study from Greece showed an incidence of any stage ROP in infants <32 weeks of 19.7% and severe ROP 7.4%.[6] The EXPRESS Study from Sweden found an incidence of any stage ROP in infants <31 weeks of 31.9% and severe ROP 5.7%.[7] There is a marked difference, however, between Indonesia and HIC in the type of infants cared for and their survival. In Indonesia, almost no infants born after a gestational age of 26 weeks or less will survive. Mortality among infants of 26–28 weeks is much higher than in HIC. In our survey, we found that 67% of infants born <28 weeks died. Recent data from Sweden, England, France, the Netherlands, Canada and the USA show a survival rate for infants born between 26 and 28 weeks of 80%–90%.[8–18] In Indonesia, only the healthier very preterm infants will survive. The incidence of ROP may be lower in these infants than in the very sick newborns who died. The same may be true of infants born after 28–32 weeks. In Indonesia, we found that 24% of these infants died. In HIC, that figure is less than 5%. The present data on the incidence of ROP in Indonesia could be an underestimation caused by the higher death rate in Indonesia. There are more reasons why our data

might be an underestimation of the real incidence of ROP in Indonesia. Not all hospitals in Indonesia have an ophthalmologist, and therefore not all preterm infants are screened. Screening might not be according to the recommended schedule in all infants so that ROP can be missed even in screened infants. Infants might be too sick to be screened, and infants might not be screened after discharge.

Studies conducted in other LMIC up to 2015 also showed a higher incidence of ROP than in HIC. In addition, ROP was seen in infants with a higher gestational age and birth weight. A study from the Philippines showed a ROP incidence of 14% in all infants born before 36 weeks.[19] A small study from Brunei showed a prevalence of 35% in infants with a birth weight of 1300±500 g and a gestational age of 29.5±2.6 weeks.[20] In Thailand, a ROP incidence of 14% was found in infants with a mean birth weight of 1514 g and a gestational age of 31.8 weeks.[21] In India, ROP has been reported to occur in 21.7%–51.9% of low birthweight infants. Most studies reported the mean birth weight of babies developing ROP to be above 1250 g and the incidence of severe ROP ranging from 5.0%–44.9%.[22] In line with our findings for the period 2005–2015, these data indicate that ROP is prevalent in LMIC, including in infants with a higher birth weight and gestational age. A recent paper describes the current state of ROP in eight LMICs.[23] The incidence of ROP was not available for all countries. This incidence, mostly based on smaller studies in one institution, ranged from 14% to 50%. In almost all countries, infants up to 34 weeks and with a birth weight of 2000 g were screened. A study from Thailand, where only infants born <30 weeks and with a birth weight of <1500 g were screened, found an ROP incidence of 40%. In all countries, the screening rate was low, at <35%. The reasons mentioned for the high incidence of ROP was similar for all countries: a lack of awareness among paediatricians, a shortage of trained ophthalmologists and a lack of funds for screening. Almost all countries lacked oxygen delivery systems and oxygen saturation monitors. All countries fear an epidemic of blind infants as a result of ROP. In our view, the results of our survey indicate that it is possible to reduce the incidence of ROP, also in LMIC. The first step to stop this epidemic is to be aware of the risks of ROP. This concerns all those involved in the care of preterm infants, paediatricians, ophthalmologists, nurses and administrators.

Our study found that the screening rate for ROP is rather low in Indonesia, both in infants with a low gestational age and in infants with a higher gestational age who received supplemental oxygen for a prolonged period. This is most likely due to at least three factors: a lack of trained ophthalmologists, a lack of awareness among paediatricians of the importance of screening and a lack of funding for ophthalmologists. Paediatric ophthalmologists are mainly found in large academic hospitals and the national centres for perinatology. In almost all cases, there is only one paediatric ophthalmologist who is not always available. In order to increase the screening rate,

paediatricians must be made aware of its importance and ophthalmologists must be trained to do it. Funds to carry out the screening need to be made available. It will not be possible to have, in a short period, enough trained ophthalmologists in Indonesia to have all preterm infants requiring ROP screening, screened according to the international accepted screening protocols. A potential solution to the lack of trained ophthalmologists might be cameras to make images of the retina and have these images evaluated by qualified, non-medical personnel. These assistants can send pictures of infants who might need ROP treatment via the internet to trained ophthalmologists. Simple, not expensive cameras have been developed. This system's advantage is that time required from ophthalmologists is reduced, and pictures can also be made in smaller hospitals without a trained ophthalmologist. This system is now implemented in India's parts, where it has been shown to be very effective. A sensitivity of 98% is achieved in detecting ROP cases that need intervention.[24 25]

We found no difference in the incidence of ROP in the institutions with a screening rate of at least 80%, compared with the whole group of infants. This indicates that a high number of infants are not screened while they develop ROP. The follow-up of preterm infants admitted to a NICU is low in Indonesia because of socioeconomic factors. It is not therefore known how often ROP was present in surviving, unscreened infants.

A lack of ophthalmologists trained in ROP screening is not a problem that is unique to Indonesia. There is a shortage of trained ophthalmologists in many LMIC and, where they do exist, screening is often not properly reimbursed and only the most committed ophthalmologists are willing to screen.[26]

In addition, paediatricians participating in this survey reported a reluctance among ophthalmologists to screen because of medico-legal problems, an imbalance between the level of difficulty and time spent examining very small premature babies and the results obtained, and the asynchronous examination fees in different health insurance systems. This all contrasts with the practice in HIC, where failure to screen an eligible infant for ROP could be considered malpractice.

The incidence of ROP in Harapan Kita Women and Children's Hospital was significantly lower in the period 2016–2017 than in the period 2005–2015. At the same time, the screening rate improved to 93%. This indicates that ROP declined significantly at that hospital. In 2016, a strict policy was introduced to set the oxygen saturation monitor at 91%–95% for preterm infants. The use of oxygen in the delivery room was also strictly regulated, with resuscitation of preterm infants starting at 30% oxygen. Continuous positive airway pressure was given directly after birth and continued in the NICU. These measures are likely to have played an important role in reducing the incidence of ROP at that NICU.

The highest rate of both mortality and ROP was found in the university-based hospitals. These are referral

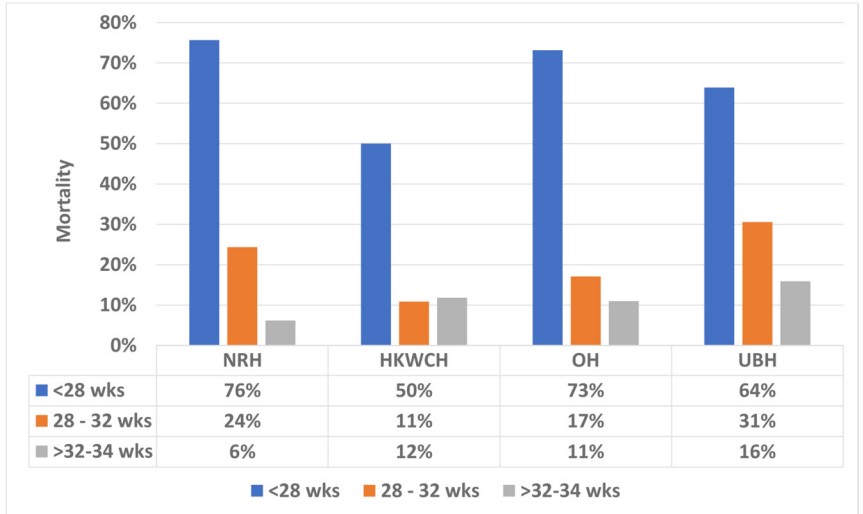

**Figure 2** Mortality based on the gestational age in each group of hospitals in Indonesia, 2016–2017. HKWCH: Harapan Kita Women and Children's Hospital (Health Centre); NRH, national referral hospital; OH, other hospitals; UBH, university-based hospital; wks, weeks.

hospitals that admit mothers with antenatal complications. They may therefore treat the sickest infants, sicker than those cared for in governmental and private hospitals. This could explain the higher rate of mortality and ROP. At the same time, however, the two national centres are also referral hospitals and they showed a lower mortality rate and a lower incidence of ROP (figures 1 and 2). This suggests that there could be other factors behind the higher incidence of ROP in the university-based hospitals. More liberal use of oxygen might be a cause. More studies are needed to determine the cause of the less favourable outcome in the university-based hospitals.

A limitation of this study is that the survey on the incidence of ROP as carried out in this study had not been done before. As a reference in this paper, we used data from one referral hospital as well as data from the literature. The rates of ROP found in one hospital in Indonesia[1] and the studies done in other centres[2] all showed a high incidence of ROP in the period before 2015. Therefore, we are convinced that the incidence of ROP was indeed much higher than suggested by the results of this survey. A second limitation of our survey is that we only included inborn infants. We do not have precise data on the ratio of inborn and outborn infants for all hospitals. In the Harapan Kita Hospital, one of the national referral hospitals, on average 71% of the admitted infants are inborn. We estimate that this percentage will be almost the same for the university-based NICUs. Unfortunately, there is no adequate neonatal transport service in Indonesia. Transportation is carried out by poorly trained personnel and only 100% oxygen can be given during transport. The referring and accepting neonatologists do not meet, making the transfer of information difficult and often incomplete.

## CONCLUSION

The incidence of ROP in preterm infants in Indonesia was lower in 2016–2017 than in the period before 2015. This may be due to a higher awareness of ROP among practitioners and the introduction of a national healthcare plan. Our data, however, are likely to be an underestimation of the real incidence of ROP because of the higher mortality rate among small premature infants in Indonesia than in HIC and a low rate of screening and follow-up of surviving preterm infants.

**Author affiliations**
[1]Neonatology, Universitair Medisch Centrum Groningen, Groningen, The Netherlands
[2]Neonatology, Harapan Kita National Centre for Women and Children's Health, Jakarta, Indonesia
[3]Neonatology, University Medical Center Groningen Intensive Care Medicine, Groningen, The Netherlands
[4]Pediatric, University of Indonesia Faculty of Medicine, Jakarta, Indonesia
[5]Pediatric, Dr Kariadi General Hospital Medical Center, Semarang, Central Java, Indonesia
[6]Pediatric, Dr Saiful Anwar General Hospital, Malang, Jawa Timur, Indonesia
[7]Pediatric, University of Sumatera Utara Faculty of Medicine, Medan, North Sumatera, Indonesia
[8]Pediatric, University of Riau Faculty of Medicine, Pekanbaru, Riau, Indonesia
[9]Pediatric, Sam Ratulangi University Faculty of Medicine, Manado, North Sulawesi, Indonesia

**Acknowledgements** The authors would like to thank all infants and families whose data contributed to the IMSROP Study Group. We wish to thank our ophthalmologists Dr Habsyiyah, Dr Nani H. Widodo and Dr Florence Manurung. Together with all the ophthalmologists at each hospital included in this study, they were involved in examining patients and providing scientific input in the field of ophthalmology. We would also like to thank Professor Rita S Sitorus, who encouraged us to work together to reduce the incidence of ROP in Indonesia in terms of neonatology, ophthalmology and public health. We also thank all collaborators who are members of the IMSROP Study Group (Indonesian Multicenter Retinopathy of Prematurity) for providing data in this survey so that the entire data can be combined for analysis.

**Collaborators** *Members of the IMSROP (Indonesian Multicenter Retinopathy of Prematurity) Study Group: Rinawati Rohsiswatmo (Rumah Sakit Cipto

Mangunkusumo, Jakarta Pusat), Tetty Yuniati (RSUP Hasan Sadikin, Bandung, Jawa Barat), Gatot Irawan Sarosa (RSUP Dr Kariadi, Semarang, Jawa Tengah), Dwi Hidayah (RSUD Dr Moewardi, Solo, Jawa Tengah), Dina Angelika, Risa Etika (RSUD Dr Soetomo Surabaya, Jawa Timur), Eko Sulistijono (RSUD Dr Syaiful Anwar Malang, Jawa Timur), Made Kardana (RSUP Sanglah, Denpasar, Bali), Pertin Sianturi (RSUP H. Adam Malik, Medan, Sumatera Utara), Eni Yantri (RSUP Dr M Djamil Padang, Sumatera Barat), Dewi Anggraini Wisnumurti (RSUD Arifin Achmad Pekanbaru, Riau), Afifa Ramadanti (RSUP M Hoesin Palembang, Sumatera Selatan), Pudji Andayani (RSUD Ulin Banjarmasin Kalimantan Selatan), Rocky Wilar (RSUP Professor Dr RD Kandou Hospital, Manado), Ema Alasiry (RSUP Dr Wahidin Sudirohusodo, Makasar, Sulawesi Selatan), Harris Alfan (RS Atma Jaya, Jakarta Utara), Johanes Edy Siswanto (RSAB Harapan Kita, RS Royal Taruma, Jakarta Barat, FK Universitas Pelita Harapan, Tangerang), Naomi Esthernita (Siloam Hospital Kebon Jeruk, Jakarta Barat), Fahrul Wakil Arbi (RS Budi Kemuliaan, Jakarta Pusat), Nita Dewanti (RS Premier Bintaro, Jakarta Selatan), Agnes Yuni Purwita Sari (RSUP Persahabatan, Jakarta Timur), Ellen Sianipar (RSUD Pasar Rebo Jakarta Timur), Robert Soetandio (RS Awal Bros Hospital, Tangerang, Jawa Barat), Andika Tiurmaida (RSUD Cibinong, Jawa Barat), Thomas Harry Adoe (RSUD Kota Bekasi), Muda Isa Ariantana (Santosa Hospital Bandung Cermai Bandung, Jawa Barat), Samad Suparman (RSUP Dr Soeradji Tirtonegoro Klaten), Anince Kwelim (RS Bakti Timah Karimun, Kepulauan Riau), Tri Putri Yuliani (RSUD Siti Aisyah Kota Lubuk Linggau), Sumsel, Dina Frida (RSUD Dr Sudarso, RS Anugrah Bunda Khatulistiwa, Kalimantan Barat), Nurhandini Eka Dewi (RSUD Mataram, Nusa Tenggara Barat), Woro Indri P Purba (RSUD Professor Dr WZ Johannes Kupang, Nusa Tenggara Timur), Sadiyah Manda Tikupadang (RSUD I Lagaligo, Luwu Timur, Sulawesi Selatan).

**Contributors** JES is the person directly involved in all aspects of this research and manuscript. JES, AFB, PHD and PJJS contributed to the study design and analysis. PJJS assisted in the literature review and wrote the report and the discussion section. AFB and PJJS are the guarantors for the study. JES, AFB and PJJS had full access to the IMSROP data. All authors had full access to the summary data presented in this paper (including statistical reports and tables) and take responsibility for the integrity of the data and the accuracy of the data analysis.

**Funding** The authors have not declared a specific grant for this research from any funding agency in the public, commercial or not-for-profit sectors.

**Competing interests** None declared.

**Patient consent for publication** Not required.

**Ethics approval** The study was approved by the Institutional Review Board of the University of Indonesia School of Public Health (No. 32/H2.F10/PPM.00/2013).

**Provenance and peer review** Not commissioned; externally peer reviewed.

**Data availability statement** All data relevant to the study are included in the article or uploaded as supplemental information. No additional information.

**ORCID iD**
J Edy Siswanto http://orcid.org/0000-0002-6209-9312

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
