## [Reviewer comments · BMJ Paediatrics Open]

ARTICLE DETAILS

TITLE (PROVISIONAL)	A multicentre survey of retinopathy of prematurity in Indonesia
AUTHORS	Siswanto, J E Bos, Arend F. Dijk, Peter H Rohsiswatmo, Rinawati Irawan, Gatot Sulistijono, Eko Sianturi, Pertin Wisnumurti, Dewi A Wilar, Rocky Sauer, Pieter JJ Study Group, The IMSROP

VERSION 1 – REVIEW

REVIEWER	Reviewer name: Dr. Brian Darlow Institution and Country: University of Otago Christchurch, Paediatrics, United Kingdom of Great Britain and Northern Ireland Competing interests: None
REVIEW RETURNED	23-Jun-2020

GENERAL COMMENTS	What has been characterised as the third epidemic of ROP has particularly affected countries in Southeast Asia. It is clear that many cases of visual impairment from ROP are preventable by improved neonatal care, timely retinal examination and appropriate treatment (see eg Darlow B et al. Clin Perinatol 2013;40:215-27). But there are few reports on the implementation and effectiveness of such measures by regions or countries hence the present report, which attempts to do this for Indonesia, is to be welcomed. The method used is a survey questionnaire covering the years 2016-7. It is stated that data were collected from all levels of hospitals where care for sick newborn infants is given: 34 hospitals were invited to participate. Indonesia has a very large population spread over many islands and so it seems surprising that 34 constitutes all hospitals providing such care. Please clarify. Please clarify also whether this was a prospective or retrospective survey. Please clarify that data are reported for only inborn infants. Is that true also for the two “national referral centers for perinatology”? We are also informed in the Discussion that the University-based hospitals are referral hospitals so, again, please confirm (if true) that only inborn infants are included in this survey. Please define “severe ROP” in the Methods. (I think probably this stage 3 or more, but other definitions are frequent in the literature). According to local guidelines, only 37% of eligible infants were examined for ROP, and only 63% of those with gestation <28 weeks. This low rate of screening is briefly mentioned in the
--

	Discussion in the context of possible underestimation of the incidence of ROP but further discussion of the reasons for, and possible solutions to, the problem are warranted. In high income countries, failure to screen an eligible infant for ROP appropriately could be considered malpractice. Limitations of the study include that this is a retrospective survey (if true) and concerns only inborn infants. It might be helpful to have some comments about the proportions of infants these hospitals care for that are outborn because they are likely to be at higher risk of any and severe ROP. Other comments. Page 5, line 41 "In order to evaluate whether the attention....." does not make sense in English. Page 10, line 39 "resuscitation....was done with in principle 30% oxygen". I take it this means "commencing with 30% oxygen". In Tables, please give percentages to only one decimal point. The text refers to the incidence of ROP whereas the Tables/Figures use prevalence. I suggest incidence is correct. The Abstract should note that is a retrospective survey (if true).
--	--

REVIEWER	Reviewer name: Dr. Ismail Shatriah Institution and Country: N/A Competing interests: None
REVIEW RETURNED	02-Jul-2020

GENERAL COMMENTS	General comments This manuscript requires an extensive English editing. 1) Title - Please rephrase the title. Avoid comma and abbreviation if possible. 2) Abstract - The content is sound but require a proper english editing servive. 3) Introduction - The implementation of national guideline was not written clearly (refer page 5, first paragraph) - Elaborate about marked variation between the options....(refer page 5, line 33) 4) Methods - I'm a bit confuse. The data asked from other hospitals appeared as audit/clinical data and not a proper questionnaire - Design of the study should be mentioned clearly - It is important to state clearly the duration of the study. Were all data were confined to postnatal period? - The abbreviations (e.g. NICU) need to spell out for the first time use. Refer page 5,line 30) 5) Results - The results are described in frequency and percentage. No advance statistical test was used for data analysis 6) Discussion - the discussion is fair
---

	- however, it will be more informative if the authors made a parallel comparison with other studies from developing countries especially from Asia
--	--

VERSION 1 – AUTHOR RESPONSE

Dear editor,
Dr. Karel Allegaert
Associate Editor, BMJ Paediatrics Open

Prof. Imti Choonara
Editor in Chief, BMJ Paediatrics Open

Thank you very much for the review of our Manuscript ID bmjpo-2020-000761, entitled "Indonesian Multicenter Study of ROP, a survey in 34 hospitals." We have made the changes as suggested by the reviewers, we feel we could answer all issues raised.

We have also sent the revised manuscript to the language institute in Groningen in order to get a full improvement in terms of English grammar. And send before the deadline set by the editor.

We send the responses in the form of a file.

Also, the manuscript that we have highlighted.

In addition, we also convey that the number of words in this type of article exceeds that which has been determined because we add and change a number of words, sentences, or paragraphs suggested by reviewers and language institutions that provide corrections to improve this manuscript.

Yours Sincerely,
J. Edy Siswanto
On behalf of all the authors

Reviewer: 1

Comments to the Author

What has been characterised as the third epidemic of ROP has particularly affected countries in Southeast Asia. It is clear that many cases of visual impairment from ROP are preventable by improved neonatal care, timely retinal examination and appropriate treatment (see eg Darlow B et al. Clin Perinatol 2013;40:215-27). But there are few reports on the implementation and effectiveness of such measures by regions or countries hence the present report, which attempts to do this for Indonesia, is to be welcomed.

Dear Prof. Brian Darlow,

Thank you very much for taking the time to provide a review of our manuscript. We also thank you for your positive words. We agree that it is important to explain the incidence and implementation of the ROP screening strategy in Indonesia, as part of the Southeast Asian region.

The method used is a survey questionnaire covering the years 2016-7. It is stated that data were collected from all levels of hospitals where care for sick newborn infants is given: 34 hospitals were invited to participate. Indonesia has a very large population spread over many islands and so it seems surprising that 34 constitutes all hospitals providing such care. Please clarify. Please clarify also whether this was a prospective or retrospective survey.

In MS we write it like below,

“This is a retrospective survey: we collected data for the years 2016-2017 in the period from March to November 2019. Paediatricians in 47 hospitals were contacted by email and direct phone calls; 41 were willing to send us the required information. We received responses from 34 hospitals in 17 major provinces of Indonesia – 16 teaching hospitals, two of which are national referral hospitals for perinatology, and 10 government and eight private hospitals. The availability of NICU beds varied greatly across between regions because of a lack of trained neonatologists and differences in stakeholder support in the province or district where the paediatricians worked. We approached hospitals offering all levels of neonatal care, located in all the different parts of Indonesia.”

Please clarify that data are reported for only inborn infants. Is that true also for the two “national referral centers for perinatology”? We are also informed in the Discussion that the University-based hospitals are referral hospitals so, again, please confirm (if true) that only inborn infants are included in this survey.

The data we requested and collected is indeed the data of new babies born at the hospital (inborn babies). We excluded all babies born outside the hospital (outborn babies). Likewise, we applied the request to 2 national referral centers and other university-based hospitals.

We added to the method section: “We included only inborn infants because important data such as gestational age and complications in pregnancy are often not available for outborn infants.”

Please define “severe ROP” in the Methods. (I think probably this stage 3 or more, but other definitions are frequent in the literature).

That's right, the definition we use for severe ROP in this paper is ROP stage 3 or higher, and we will write it in the method

In the method section we added,

“For the sake of uniformity and to analyse all the data, we asked for inborn babies to be further categorised by gestational age and birth weight. For ROP, we used the terms mild ROP (stages 1-2) and severe ROP (stage 3 or higher).”

According to local guidelines, only 37% of eligible infants were examined for ROP, and only 63% of those with gestation <28 weeks. This low rate of screening is briefly mentioned in the Discussion in the context of possible underestimation of the incidence of ROP but further discussion of the reasons for, and possible solutions to, the problem are warranted. In high income countries, failure to screen an eligible infant for ROP appropriately could be considered malpractice.

We fully agree with the reviewer that the rate of screening is very low in our population. This is most likely due to at least three factors, a lack of trained ophthalmologists, a lack of awareness of the importance of screening by pediatricians and a lack of funds for ophthalmologists.

We added in the discussion the following.

“Our study found that the screening rate for ROP is rather low in Indonesia, both in infants with a low gestational age and in infants with a higher gestational age who received supplemental oxygen for a prolonged period. This is most likely due to at least three factors: a lack of trained ophthalmologists, a lack of awareness among paediatricians of the importance of screening and a lack of funding for ophthalmologists. Paediatric ophthalmologists are mainly found in large academic hospitals and the national centres for perinatology. In almost all cases, there is only one paediatric ophthalmologist who is not always available. In order to increase the screening rate, paediatricians must be made aware of its importance and ophthalmologists must be trained to do it. Funds to carry out the screening need to be made available.”

Limitations of the study include that this is a retrospective survey (if true) and concerns only inborn infants. It might be helpful to have some comments about the proportions of infants these hospitals care for that are outborn because they are likely to be at higher risk of any and severe ROP.

We agree with reviewer 1 that the limitation of this study is that the data obtained from the results of a retrospective survey and concerns only inborn infants.

As per the reviewer's suggestion, in our MS, we add in the paragraphs of the study limitations as below,

"A second limitation of our survey is that we only included inborn infants. We do not have precise data on the ratio of inborn and outborn infants for all hospitals. In the Harapan Kita Hospital, one of the national referral hospitals, on average 71% of the admitted infants are inborn. We estimate that this percentage will be almost the same for the university-based NICUs. Unfortunately, there is no adequate neonatal transport service in Indonesia. Transportation is carried out by poorly trained personnel and only 100% oxygen can be given during transport. The referring and accepting neonatologists do not meet, making the transfer of information difficult and often incomplete."

Other comments.

Page 5, line 41 "In order to evaluate whether the attention ..." does not make sense in English.

Thank you for the correction, we have written the fix as below,

To evaluate whether awareness of the high incidence of ROP in Indonesia and the introduction of the national health insurance may have reduced the incidence of ROP in Indonesia, we

Page 10, line 39 "resuscitation is done with the principle of 30% oxygen". I take it this means "commencing with 30% oxygen".

Thank you for the correction, we have fixed it

with resuscitation of preterm infants starting at 30% oxygen.

In Tables, please give percentages to only one decimal point.

Thank you for the correction, we have fixed it in the table

The text refers to the incidence of ROP whereas the Tables / Figures use prevalence. I suggest incidence is correct.

Thank you for the correction, we have replaced the word prevalence with incidence

The Abstract should note that it is a retrospective survey (if true).

Thank you, we have entered the word retrospective survey in Abstract

Reviewer: 2

 Comments to the Author

General comments

This manuscript requires an extensive English editing.

Dear Professor Dr. Shatriah Ismail

Thank you for your time to review this text, also for all your constructive comments, we will improve it through a competent English language institution to make this MS better.

1) Title

- Please rephrase the title. Avoid comma and abbreviation if possible.

Thank you for the input, we have improved the title by replacing it as suggested by the Chief Editor.

"A multicenter survey of retinopathy of prematurity in Indonesia".

2) Abstract

- The content is sound but requires a proper English editing service.

Thank you for reviewer 2's comments, for the content written in our paper, we have edited it in English in the manuscript according to Prof.'s advice.

3) Introduction

- The implementation of national guidelines was not written clearly (refer page 5, first paragraph)

We have fixed it by replacing it with a sentence like the one below,

"Neonatologists and paediatric ophthalmologists realized that the guideline's publication had almost no impact on clinical practice. This was most likely due to a lack of knowledge about ROP among paediatricians and ophthalmologists and to financial constraints in hospitals."

- Elaborate about marked variations between the options (refer page 5, line 33)

We have fixed it by adding it as below,

"There are marked variations among these hospitals, such as patients' socioeconomic and cultural backgrounds and other demographic factors, in terms of their options for caring for sick preterm infants. We do not know whether the differences between hospitals resulted in a different incidence of ROP."

4) Methods

- I'm a bit confuse. The data asked from other hospitals appeared as an audit / clinical data and not a proper questionnaire

We apologize for the confusion that might have been caused by the word "questionnaire". We did not send a questionnaire to the hospitals, but asked them to fill in forms where all information needed for the survey could be given. In contact with the pediatricians, we made it very clear that this study was not an audit, data for individual hospitals would not be reported, except for the two national referral centers. We deleted the word questionnaire from the manuscript.

- Design of the study should be mentioned clearly

We added that this is a retrospective survey

- It is important to state clearly the duration of the study. Were all the data were confined to postnatal period?

- We have written in the method, as we explained in the sentence below,

“This is a retrospective survey: we collected data for the years 2016-2017 in the period from March to November 2019. Paediatricians in 47 hospitals were contacted by email and direct phone calls; 41 were willing to send us the required information. We received responses from 34 hospitals in 17 major provinces of Indonesia – 16 teaching hospitals, two of which are national referral hospitals for perinatology, and 10 government and eight private hospitals. The availability of NICU beds varied greatly across between regions because of a lack of trained neonatologists and differences in stakeholder support in the province or district where the paediatricians worked. We approached hospitals offering all levels of neonatal care, located in all the different parts of Indonesia.”

- Yes all data is limited to the postnatal period

- The abbreviations (e.g. NICU) need to spell out for the first time use. Refer to page 5, line 30)

Thank you, we have changed it NICU (Neonatal Intensive Care Unit)

5) Results

- The results are described in frequency and percentage. No advance statistical test was used for data analysis

Thank you for the input, we have added this information in the Methods section.

6) Discussion

- the discussion is fair

- However, it would be more informative if the authors made a parallel comparison with other studies from developing countries, especially from Asia

Thank you, we have added it like the sentence below,

“Studies conducted in other LMIC up to 2015 also showed a higher incidence of ROP than in HIC. In addition, ROP was seen in infants with a higher gestational age and birth weight. A study from the Philippines showed a ROP incidence of 14% in all infants born before 36 weeks.¹⁹ A small study from Brunei showed a prevalence of 35% in infants with a birth weight of 1300 ± 500 g and a gestational age of 29.5 ± 2.6 weeks.²⁰ In Thailand, a ROP incidence of 14% was found in infants with a mean birthweight of 1514 g and a gestational age of 31.8 weeks.²¹ In line with our findings for the period 2005-2015, these data indicate that ROP is prevalent in LMIC, including in infants with a higher birthweight and gestational age. A recent paper describes the current state of ROP in eight LMIC.²² The incidence of ROP was not available for all countries. This incidence, mostly based on smaller studies in one institution, ranged from 14% to 50%. In almost all countries, infants up to 34 weeks and with a birthweight of 2000 g were screened. A study from Thailand, where only infants born <30 weeks and with a birthweight of <1500 g were screened, found a ROP incidence of 40%. In all countries, the screening rate was low, at <35%. The reasons mentioned for the high incidence of ROP was similar for all countries: a lack of awareness among paediatricians, a shortage of trained ophthalmologists and a lack of funds for screening. Almost all countries lacked oxygen delivery

systems and oxygen saturation monitors. All countries fear an epidemic of blind infants as a result of ROP. In our view, the results of our survey indicate that it is possible to reduce the incidence of ROP, also in LMIC. The first step to stop this epidemic is to be aware of the risks of ROP. This concerns all those involved in the care of preterm infants, paediatricians, ophthalmologists, nurses and administrators."

Editor in Chief

Comments to the Author:

Add your questionnaire as an appendix.

We apologize for the confusion that might have been caused by the word "questionnaire". We did not send a questionnaire to the hospitals, but asked them to fill in forms where all information needed for the survey.

Yes we will add our ROP "survey data form" as an appendix.

Title- amend to "A national survey of retinopathy of prematurity in Indonesia". If it is NOT national, replace national with multicenter

Thank you for the input, we have changed the title to "A multicenter survey of retinopathy of prematurity in Indonesia".

Clarify whether you asked more than 34 hospitals to participate, and if so how many. Are these 34 hospitals throughout Indonesia?

Yes, all 34 participating hospitals are throughout Indonesia. We invited 47 neonatologist (respondents) via email and telephone directly to discuss the possibility of their participation in the adoption of the agreed ROP screening guidelines. We got the desire to participate from 41 pediatricians, and we received responses from filling out survey sheets from 34 hospitals in 17 major provinces in Indonesia, 16 teaching hospitals and 2 of them were national referral hospitals for perinatology. The most common reason for not participating was the absence of a trained ophthalmologist at the prospective respondent's hospital so that the pediatrician sent premature infants in eligible populations to hospitals that had service facilities for ROP screening, in addition to the ophthalmologist's reluctance to examine small premature babies because the level of difficulty and requires a long time.

Results text states the other 18 hospitals included 11 government hospitals, whereas the table states it is 10.

Thank you for your correction. We have fixed it.

The correct data is 10 government hospitals and 8 private hospitals

Table 1. The number of babies and survivors for the government & private hospitals does not equal the total for Other hospitals. Please correct.

Yes we have fixed it directly in table 1

Table 2 is too big. Make one table of 2005-2015 data and 2016-2017 data from HKWCH to allow direct comparison. The other table should include 2016-2017 data only.

Yes, we have divided table 2 into two different tables namely

Table 2a. ROP incidence in Harapan Kita Women and Children Hospital, Indonesia based on the gestational age in 2005-2017

Table 2b. ROP incidence in Indonesia based on the gestational age in 2016-2017

Table 3 adds a footnote that the data are from 13 out of the 34 hospitals

Yes we have added it as a footnote in table 3

Results Would be easier to read if you write one paragraph about each table.

Yes we have changed and improved the paragraph

Avoid use of "developing / developed" countries. Use high income / lower middle instead.

Yes we have replaced the use of the word "developing / developed" countries to high-income /lower-middle countries

Discussion focus on studies from lower middle income countries only (not high income)

We have added it to the discussion paragraph comparing the conditions of ROP screening and management in Indonesia with countries in Southeast Asia and other parts of Asia.

Discussion do NOT repeat results (see first four sentences). Your paper needs a major rewrite, especially with regards to the English. If you need more time, please let us know

Yes, we immediately corrected and sent it to the language institute to be able to correct the grammar we used. For this step we may need more time to wait for this process to finish.

VERSION 2 – REVIEW

REVIEWER	Reviewer name: Dr. Ismail Shatriah Institution and Country: N/A Competing interests:
REVIEW RETURNED	07-Sep-2020

GENERAL COMMENTS	The authors have listed two causes of low incidence of ROP in Indonesia. Any other possible causes contributing to this outcome e.g. clinical competency? It will be more meaningful if the comparisons are made with data from Asian countries and developed countries.
---

REVIEWER	Reviewer name: Dr. Brian Darlow Institution and Country: University of Otago Christchurch, Paediatrics, United Kingdom of Great Britain and Northern Ireland Competing interests: None
REVIEW RETURNED	24-Aug-2020

GENERAL COMMENTS	This is a resubmission. The authors have satisfactorily dealt with all the issues raised by this reviewer. Given the lack of ophthalmologists available or willing to undertake ROP examinations in Indonesia, the authors might consider adding a brief comment on the alternative approach of using digital imaging undertaken by trained non-ophthalmologists (eg neonatologists, nurses, technicians) with remote reading of the images. See Gilbert et al 2016 for an overview and Quinn and Anand 2019, which gives information on the experience in India where digital images are often taken with newer handheld cameras or even cell-phones. Gilbert C, Wormald R, Fielder A, et al. Potential for a paradigm change in the detection of retinopathy of prematurity requiring treatment. Arch Dis Child Fetal Neonatal Ed 2016; 101: F6-9. Quinn GE, Vinekar A. The role of retinal photography and telemedicine in ROP screening. Semin Perinatol. 2019; 43(6): 367-374.
---

VERSION 2 – AUTHOR RESPONSE

Dear,
Prof. Imti Choonara
Editor in Chief, BMJ Paediatrics Open

Dr. Karel Allegaert
Associate Editor, BMJ Paediatrics Open

We thank you for your mail indicating that we need to make some minor changes in our paper to have it accepted. In answer to your comment, we deleted the word "retrospective" in the abstract. Our responses to the reviewers are attached. We hope the paper can now be accepted.

Answer to Reviewer 1 (Prof. Dr. Brian Darlow)

We thank the reviewer for this excellent suggestion. We added to the paper the following.

It will not be possible to have, in a short period, enough trained ophthalmologists in Indonesia to have all preterm infants requiring ROP screening, screened according to the international accepted screening protocols. A potential solution to the lack of trained ophthalmologists might be cameras to make images of the retina and have these images evaluated by qualified, non-medical personnel. These assistants can send pictures of infants who might need ROP treatment via the internet to trained ophthalmologists. Simple, not expensive cameras have been developed. This system's advantage is that time required from ophthalmologists is reduced, and pictures can also be made in smaller hospitals without a trained ophthalmologist. This system is now implemented in India's parts, where it has been shown to be very effective. A sensitivity of 98% is achieved in detecting ROP cases that need intervention. (Gilbert 2016, Quinn 2019)

Answer to Reviewer 2 (Prof. Dr. Ismail Shatriah)

We thank the reviewer for the comments. It is correct that the prevalence of ROP as found by us in this study might be an underestimation. There are indeed more reasons for an underestimation than written in the present manuscript. We, therefore, added the following to the paper.

There are more reasons why our data might be an underestimation of the real incidence of ROP in Indonesia. Not all hospitals in Indonesia have an ophthalmologist, and therefore, not all preterm infants are screened. Screening might not be according to the recommended schedule in all infants so that ROP can be missed even in screened infants. Infants might be too sick to be screened, and infants might not be screened after discharge.

To add a comparison between our data and data from another LMIC, we added the following to the manuscript.

In India, ROP has been reported to occur in 21.7%– 51.9% of low birth weight infants. Most studies report the mean birth weight of babies developing ROP to be above 1250 g and the incidence of severe ROP ranging from 5.0–44.9%.

(Dogra MR, Katoch D, Dogra M. An Update on Retinopathy of Prematurity (ROP). *Indian J Pediatr* 2017;12: 930-6. DOI: 10.1007/s12098-017-2404-3. Epub 2017 Jul 4).

Yours sincerely,
On behalf of all authors,
J. Edy Siswanto